# RGTI:RESPONSE GENERATION VIA TEMPLATES INTEGRATION FOR END TO END DIALOG

## ABSTRACT

End-to-end models have achieved considerable success in task-oriented dialogue area, but suffer from the challenges of (a) poor semantic control, and (b) little interaction with auxiliary information. In this paper, we propose a novel yet simple end-to-end model for response generation via mixed templates, which can address above challenges. In our model, we retrieval candidate responses which contain abundant syntactic and sequence information by dialogue semantic information related to dialogue history. Then, we exploit candidate response attention to get templates which should be mentioned in response. Our model can integrate multi template information to guide the decoder module how to generate response better. We show that our proposed model learns useful templates information, which improves the performance of "how to say" and "what to say" in response generation. Experiments on the large-scale Multiwoz dataset demonstrate the effectiveness of our proposed model, which attain the state-of-the-art performance.

## 1 INTRODUCTION

Task-oriented dialogue is aim to help users to complete a task in specific field such as restaurant reservation, or booking film tickets. The traditional approach is to design pipeline architectures which have several modules: natural language understanding, dialogue manager and natural language generation (Wen et al., 2017). It's easy to control, but the dialogue system becomes more and more complicated. With the development of deep learning, end-to-end methods have shown hopeful results and received great attention in academic. They input user queries and generate system responses, which is relatively simple. However, the disadvantage is that end-to-end approaches are difficult to control generated results.

Task-oriented responses should have correct entities and grammatical expressions, which means solving problems of "what to say" and "how to say". If a user wants to ask for a restaurant of moderate price range, a good task-oriented dialogue system should return the response with right restaurants whose prices cannot be high or low and use the proper wording which in clear and unambiguous expression. Researches get right entities by looking up the knowledge base(KB). Sukhbaatar et al. (2015) introduce KB in the form of hidden states. Madotto et al. (2018) use attention and copy mechanism to produce words from KB. Most research on generating smooth response has been carried out in using templates. However, templates usually are designed by domain experts in advance (Walker et al., 2007b) , so it needs huge cost and is difficult to transfer different domains. Wen et al. (2015) use Semantically Controlled LSTM (SC-LSTM) to control semantic results of language generation. Su et al. (2018) propose a hierarchical architecture using linguistic patterns to improve response generation. These models though achieving good performance, suffer from the problems as templates fixed or poorly controllability of semantic .

In order to alleviate such issues, we propose an end-to-end response generation model via templates integration(RGTI), which is composed of a encoder encoding the retrieval candidate responses with the form of triple into template representation and a decoder that integrates templates to generate the target response. Instead of encoding the templates directly, we construct a hierarchical encoding structure to make the templates contain semantic information and sequence information. During the decoder phase, we exploit a mixed decoder via templates and dialogue history introducing copy mechanism to generate better response.

We empirically show that RGTI can achieve advanced performance using the triple encoder and mixed decoder. In the human-human multi-domain dialogue dataset Budzianowski et al. (2018), RGTI is able to surpass the previous state-of-the-art on automatic evaluation, which further confirms the effectiveness of our proposed encoder-decoder model.

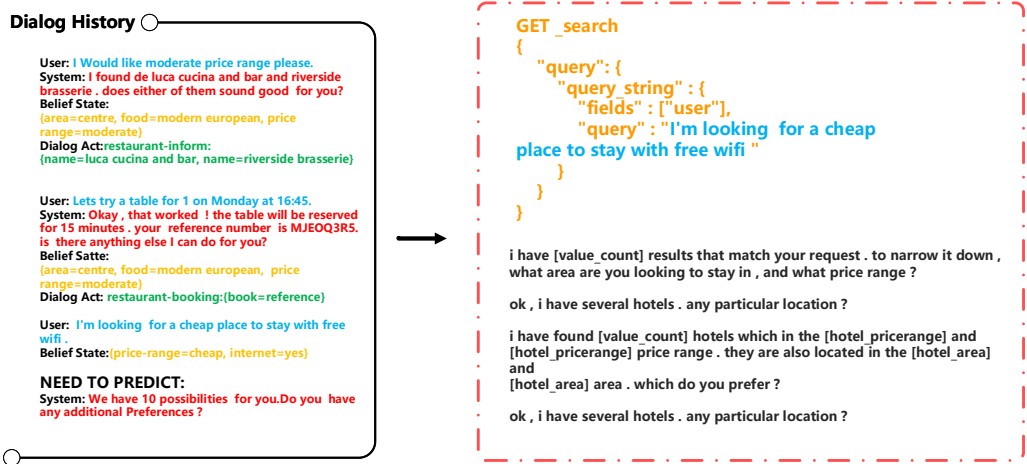

Figure 1: Dialogue Example.The left part of the figure is the raw dialogue contexts, based on the above dialogue history we obtain candidate reference responses which in the lower right part of the figure by retrieving the dialogue.

## 2 OUR MODEL

We now describe the RGTI framework composed of two parts: encoder for templates in triple form as well as dialogue history and mixed decoder in generation and copy mode, as shown in Figure 1. The dialogue history $X = (x_1, ..., x_n)$ is the input, and the system response $Y = (y_1, ..., y_m)$ is the expected output, where $n,m$ are the corresponding lengths. We first retrieval relevant candidate responses to the target responses. Then the encoder block uses multi-head attention to encode the candidate responses in triple form into templates representation. Next, the copy augmented decoder uses a gating mechanism for words selection from two sources: utterances and templates, while generating a target response.

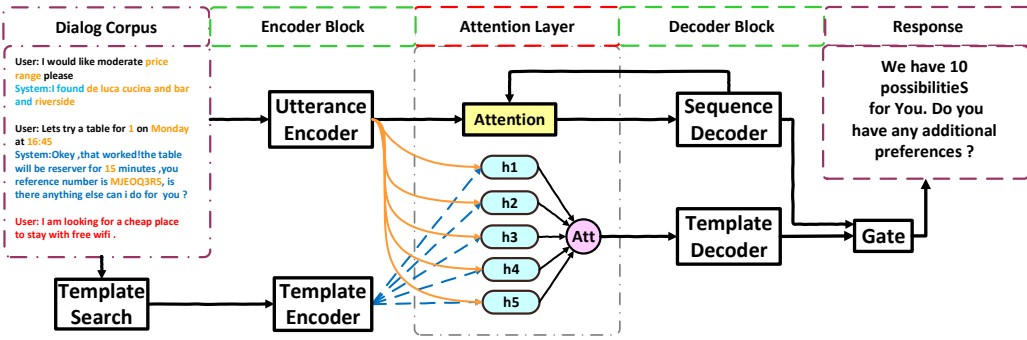

Figure 2: The proposed architecture for task-oriented dilogue systems: RGTI.

## 2.1 RETRIEVAL REFERENCE RESPONSE

First we should retrieval reference responses from training sets. The benefit of this approach is we can get more semantically coherent responses. For getting the better reference responses, we use dialogue state, dialogue act and semantic information of responses as search criteria. ElasticSearch is been used, which can get faster and more acurrate results.

## 2.2 ENCODER BLOCK

We use multi-head attention to encode dialogue history $X$ and retrieval reference responses $R$, which can automatically extract important information from sentences. If we feed the retrieved sentences directly into the model, the noise is large and not conducive to model learning, so we assume the closet part to the entity in the response is the most important. We divide the entity and its most relevant parts into triple form (head, entity, tail). Head means the part of the sentence before the entity, and tail means the after part, for example if a response is "enjoy your stay in value-place, goodbye", the corresponding head is "enjoy your stay", entity is "value-place", tail is "goodbye". Using this form, we can pay attention to word order and obtain the relationship between the sentence structure and the entity in the sentence. We treat the encoded retrieval responses as template $Q$, which is a refined expression of retrieval results.

## 2.3 MIXED DECODER

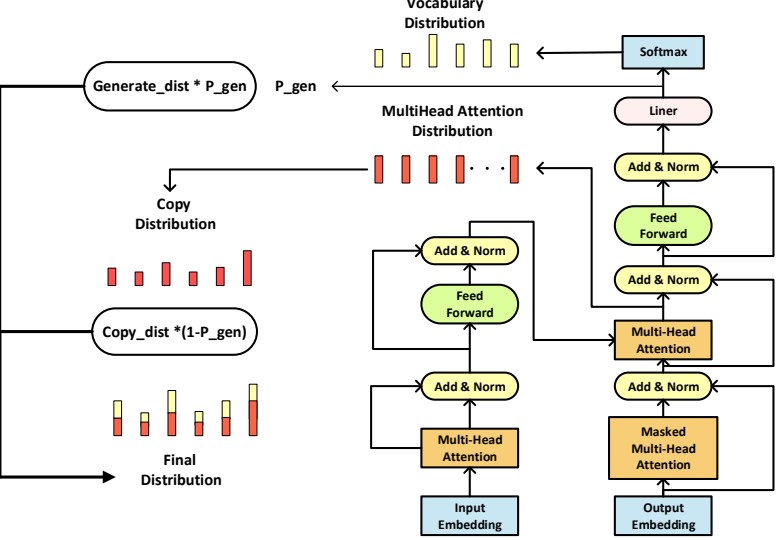

Figure 3: Mixed Decoder,

We predict tokens of target response based on two mixed modes, generate-mode and copy-mode. Generate-mode is to generate words from the vocabulary directly and copy mode is to copy words from templates. Accordingly, our model uses two output layer: sequence prediction layer, template location copying layer. Then we use the gated mechanism of the above output layers to get the final generated words. The probability of generating target word $y_t$ is the sum of multiple probabilities at each time step t:

$$p(y_t|s_t, y_{t-1}) =$$
$$p_{pr}(y_t|s_t, y_{t-1}, c_t) \cdot p_m(pr|s_t, y_{t-1}) +$$
$$p_{co}(y_t|s_t, y_{t-1}, M_Q) \cdot p_m(y_t, c_t|s_t, y_{t-1})$$

| Metric | DSTC2 | WOZ2.0 | KVRET | MultiWOZ |
|---|---|---|---|---|
| Dialogues | 1612 | 600 | 2425 | 8438 |
| Total turns | 23,354 | 4,472 | 12,732 | 113,556 |
| Total tokens | 199,431 | 50,264 | 102,077 | 1,490,615 |
| Avg turns per dialogue | 14.19 | 7.45 | 5.25 | 13.46 |
| Avg tokens per turn | 8.54 | 11.25 | 8.02 | 13.13 |
| total unique tokens | 986 | 2,142 | 2,842 | 23689 |
| Slots | 8 | 4 | 13 | 24 |
| Value | 212 | 1847 | 1363 | 4510 |

Table 1: Statistics of and MultiWOZ, part of the data above is derived from Budzianowski et al. (2018)

where $pr$ is the predict-mode, and $co$ means copy words from template, $p_m(\cdot|\cdot)$ shows the probability for choosing different modes. The probabilities of these modes are calculated as follows:

$$p_{pr}(y_t|\cdot) = \frac{1}{Z} e^{\varphi_{pr}(y_t)}$$

$$p_{co}(y_t|\cdot) = \frac{1}{Z} \sum_{j:Q_j=y_t} e^{\varphi_{co}(y_t)}$$

where $\varphi$ is score function to choose mode, Z stands for the normalization term of two modes. $Z = e^{\varphi_{pr}(v)} + \sum_{j:Q_j=v} e^{\varphi_{co}(v)}$

Specifically, the score functions of two modes are given by:

$$\varphi(y_t = v_i) = v_i^T W_{pr}[s_t, c_{tem}]$$
$$\varphi(y_t = x_j) = DNN(h_j, s_t, hist_Q)$$

## 3 EXPERIMENTS

### 3.1 DATASET

To verify the results of our model, we use recently proposed MultiWOZ dataset (Budzianowski et al., 2018) to carry out experiments. The MultiWOZ is the largest existing human-human conversational corpus spanning over seven domains (attraction, hospital, police, hotel, restaurant, taxi, train), which contains 8438 multi-turn dialogues and the average length of each dialogue is 13.68. Different from current mainstream task-oriented dialogue datasets like WOZ2.0 (Wen et al., 2017),and DSTC2 which contain less than 10 slots and only a few hundred values. There are almost 30(domain,slot) pairs and over 4500 possible values in MultiWOZ dataset. Each dialogue consists of a dialogue goal and a representation of multiple pairs of users and system utterances. In each turn of the dialogue, there are still two kinds of annotations, one is belief state and the other is the dialog action which are used to mark the status of the current conversation and the potential actions of the user.

### 3.2 TRAINING DETAILS

We trained our model end-to-end using Adam optimizer (Kingma & Ba, 2014),and uses multi-step learning rate with milestones in 50, 100, 150, 200, the learning rate starts from $1e^{-3}$ and the parameter $\gamma$ is 0.5. All the embeddings are initialized randomly, and beam-search strategy is used during the decoding state. The hyper-parameters such as hidden size is 512 and the dropout rate is 0.2.

### 3.3 EVALUATION

#### 3.3.1 ENTITY F1

The main evaluation metric is F1 score which is the harmonic mean of precision and recall at word level between the predicted answer and ground truth. By comparing the ground truth system responses with the set of entities to select useful entities, this metric can evaluate the ability to generate

| User utterance | I am interested in a restaurant that is in the expensive price range. |
|---|---|
| template1 | (is **in the**–> [restaurant_area]–> of town . shall i book you a table ?) |
| template2 | (is on the–> [restaurant_area]–> side . **would you like** me to reserve for you ?) |
| Predict | i have the [restaurant_name] **in the** [restaurant_area] . **would you like me to make a reservation ?** |
| Ground Truth | [restaurant_name] is a restaurant **in the** city [restaurant_area] with that price range . **would you like to make a reservation ?** |
| User utterance | How about one in the moderate price range? |
| template1 | (**would you prefer** the–> [hotel_area]–> **or** the) |
| template2 | (do you **prefer** the–> [hotel_area]–> **or**) |
| Predict | i have [value_count] options available . **would you prefer the** [hotel_area] **or** [hotel_area] part of town ? |
| Ground Truth | we have [value_count] entries that match your preferences . **would you prefer [hotel_area]** , [hotel_area] , **or [hotel_area]** ? |

Figure 4: Examples of RGTI generation result.

| Model | BLEU | Entity F1 |
|---|---|---|
| LSTM | | |
| Two-stage sequence-to-sequence Architectures(Sequicity) | | |
| Effectively Incorporating Knowledge Bases(Mem2Seq) | | |
| Global-to-local Memory Pointer Networks (GLMP) | | |
| Hierarchical Disentangled Self-Attention (HDSA) | | |
| Our proposed model (RGTI) | 26.89% | |

Table 2: BLEU and Entity F1 score of baselines and proposed model (RGTI) on Multiwoz dataset. Retrained using docker container provided by the authors with exactly same hyper-parameters.

relevant entities and to capture the semantics of the dialog (Eric & Manning, 2017) and (Eric et al., 2017)

### 3.3.2 BLEU

We also use BLEU score in our evaluation which is often used to compute the word overlap between the generated output and the reference response.The early BLEU metrics was used in the field of translation . In recent years, it has also referred to the BLEU metrics in the end-to-end task-oriented dialogue and in the field of chat-bots.

### 3.4 ABLATION STUDY

We perform ablation experiments on the test set to analyze the effectiveness of different module in our model. The results of these experiments are shown in the Table below. As one can observe from the Table, our model without copy mechanism has 2.5% BlEU drop in generate results. On the other hand, RGTI without templates means that we do not consider the relevant templates information and thus lead to a reduction in BLEU. Note that if we remove the templates, then a 0.5 % increase can be observed in the table, which suggests that instead of applying context history makes the corresponding text noise have more side effects than his positive influence.

## 4 RELATED WORKS

Task-oriented dialog systems are mainly explored by following two different approaches: piplines and end-to-end. For the dialogue systems of piplines (Williams Young, 2007; Wen et al., 2017), modules are separated into different trained models:(i)natural language understanding g (Young et al., 2013; Chen et al., 2016), which is used to understand human intention, dialogue state tracking

(Lee Stent, 2016; Zhong et al., 2018) which is for estimating user goal at every step of the dialogue, dialogue management (Su et al., 2016), and natural language generation (Sharma et al.,2016) which is aimed to realize language surface form given the semantic constraint. These approaches achieve good stability via combining domain-specific knowledge and slot-filling techniques, but additional human labels are needed. On the other hand, end-to-end approaches have shown promising results recently. BiLSTM Zhao et al. (2017) use recurrent neural networks to generate final responses and achieve good results the previous state-of-art model about memory mechanism strengthen the reasoning ability by incorporating external knowledge into the neural network, Sequicity and HDSA represent the conversation history as belief span or dialog action which is used as compressed information for downstream model generate system response considering current dialog state.

## 4.1 TEMPLATE

Using templates to guide response generation is a common method in task-oriented dialogue area. Stent et al. (2004) and Walker et al. (2007a) use machine learning to train the templates selection. The cost to create and maintain templates is huge, which is a challenge in adapting current dialogue system to new domains or different language. Wiseman et al. (2018) introduce HMM into text generation, which can decode the state of generation templates. It's a useful method to generate templates, however the variability and efficiency are not satisfied.

## 4.2 TRANSFORMER

Recently, a new neural architecture called transformer has surpassed RNNs on sequence to sequence tasks, the paper on transformers by Vaswani et. al. [19] demonstrated that transformers produce state-of-the-art results on machine translation, which allowing for increased parallelization and significantly reduced training time. Otherwise, there has been little work around the use of transformer on end-to-end dialog system.

## 4.3 POINTER-GENERATOR NETWORKS

The pointer-generator was first proposed by Vinyals ,and then has been applied to several natural language processing tasks, including translation (Gulcehre et al. [4]), language modeling (Merity et al. [7]), and summarization.

The motivation of this paper is how to effectively extract and use the relevant contexts for multi-turn dialogue generation. Different from previous studies, our proposed model can focus on the relevant contexts, with both long and short distant dependency relations, by using the transformer-pointer-generator mechanism.

## 5 CONCLUSION

In this paper,we propose an end-to-end trainable model called Response generation via templates Integration.The transformer pointer generator is trained to incorporate related reference responses into the learning framework which performs remarkably well. As a result our model achieves state-of-the-art results in both BLEU and Entity F1 score in the Multiwoz Dataset. We hope that the proposed model can benefit future related research.

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
