# OpenReview forum: "RGTI:Response generation via templates integration for End to End dialog"
_ICLR.cc/2020/Conference — Reject_

### Official Review · AnonReviewer3 · 2019-10-26
**Official Blind Review #3**

**Rating:** 1

**Review:**

1. Summary: The authors proposed a deep neural network-based model to generate responses fro task-oriented dialogue systems. The model mainly contains two parts, the first part is to retrieve relevant responses based on question and encode them into templates, the second part is a decoder to generate the response based on the encoded templates and input utterances.
2. Assessment: The problem studied in this work is very interesting and important. The high-level idea also looks interesting. However, this paper is clearly not ready to submit as it contains a lot of grammar errors and lacks a lot of important details.
2.1 "In our model, we retrieval ..." => "we retrieve", "which is composed of a encoder ..." => "an encoder". There are too many such language errors in this paper.
2.2 Table 2 is not finished yet. There's only one result in it.
2.3 What does "triple form" mean? What are the candidate responses to retrieve when doing inference? What are the details of reference response retrieval? To many such important details are missing.

**Experience Assessment:**

I have read many papers in this area.

**Review Assessment: Checking Correctness Of Derivations And Theory:**

I assessed the sensibility of the derivations and theory.

**Review Assessment: Checking Correctness Of Experiments:**

I assessed the sensibility of the experiments.

**Review Assessment: Thoroughness In Paper Reading:**

I made a quick assessment of this paper.

---

### Official Review · AnonReviewer2 · 2019-10-27
**Official Blind Review #2**

**Rating:** 1

**Review:**

This paper introduces a based model for the problem of task-oriented dialog. The main novelty seems to be: 1) the model can mix multiple templates to respond; 2) the response is generated using a combination of templates and a language model (directly) conditioned on the input.

The paper studies an interesting task however it is not ready for
publication:
- Empirical results of all baselines are missing (Table 2)
- Other results are mentioned but are absent from the manuscript
- The related work does not provide details as to how the proposed model to prior work
- The figures are difficult to understand (although they do help make the model clearer and so I suggest polishing them for the next version of this work)
- The manuscript should be carefully copy-edited.


**Experience Assessment:**

I have published one or two papers in this area.

**Review Assessment: Checking Correctness Of Derivations And Theory:**

N/A

**Review Assessment: Checking Correctness Of Experiments:**

I assessed the sensibility of the experiments.

**Review Assessment: Thoroughness In Paper Reading:**

I made a quick assessment of this paper.

---

### Official Review · AnonReviewer1 · 2019-10-27
**Official Blind Review #1**

**Rating:** 1

**Review:**

The basic idea of integrating templates for dialog generation is interesting. However, the implementation is confusing, yet another architecture, with clear motivations, intuitions, or even clarity. Worst of all, the experimental results are missing in table 2, the only placeholder for results in the paper. Further, I would appreciate human evaluation, rather than using BLEU as the metric. The latter is known to be inappropriate for dialog modeling evaluation.

Section 2.1 needs more explanation.

(Strong Reject)

**Experience Assessment:**

I have published one or two papers in this area.

**Review Assessment: Checking Correctness Of Derivations And Theory:**

I carefully checked the derivations and theory.

**Review Assessment: Checking Correctness Of Experiments:**

I carefully checked the experiments.

**Review Assessment: Thoroughness In Paper Reading:**

I read the paper thoroughly.

---

### Official Review · AnonReviewer4 · 2019-10-28
**Official Blind Review #4**

**Rating:** 1

**Review:**

This paper describes a method to incorporate candidate templates to aid in response generation within an end-to-end dialog system. While the motivation and task setup is interesting, the paper is clearly unfinished. Most jarringly, Table 2 which should contain the main results comparing the proposed RGTI model to existing baseline models is not filled in, and there appears to be no table showing results of the ablation study briefly described in Section 3.4.

Other issues include the lack of detail around any of the subtasks mention. For example, in Sec 2.1, how is semantic information used as search criteria to retrieve responses? We also need to see more details about where responses are retrieved from. In Sec 2.2, how is the splitting done to construct tuples from the candidate templates (the word "in" is dropped in the example given, and it's not clear if this is a typo or whether some processing is done over the phrases before tuple construction is done).

Other comments:
- The paper needs to be heavily proofread. There are many spelling/grammar mistakes, for example:
	- dilogue -> dialogue (Figure 2, page 2)
	- acurrate -> accurate (top of page 3)
        - "First we should retrieval responses" -> "retrieve responses" (page 3)
        - BlEU -> BLEU (page 5)

**Experience Assessment:**

I have published in this field for several years.

**Review Assessment: Checking Correctness Of Derivations And Theory:**

I assessed the sensibility of the derivations and theory.

**Review Assessment: Checking Correctness Of Experiments:**

I assessed the sensibility of the experiments.

**Review Assessment: Thoroughness In Paper Reading:**

I read the paper at least twice and used my best judgement in assessing the paper.

---

### Decision · Program_Chairs · 2019-12-19

**Decision:**

Reject

**Comment:**

This paper describes a method to incorporate multiple candidate templates to aid in response generation for an end-to-end dialog system. Reviewers thought the basic idea is novel and interesting. However, they also agree that the paper is far from complete, results are missing, further experiments are needed as justification, and the presentation of the paper is not very clear. Given the these feedback from the reviews, I suggest rejecting the paper.